# Short Communication: Latent Detection of Downy Mildew (*Peronospora pisi*) in Bioassays against *Pisum sativum*

**DOI:** 10.3390/pathogens10101312

**Published:** 2021-10-13

**Authors:** Alexia Marr, Seona Casonato, Eirian Jones

**Affiliations:** Department of Pest-Management and Conservation, Faculty of Agriculture & Life Sciences, Lincoln University, Lincoln 7647, New Zealand; Seona.Casonato@lincoln.ac.nz (S.C.); Eirian.Jones@lincoln.ac.nz (E.J.)

**Keywords:** *Peronospora viciae*, inoculation, latency, disease expression, peas

## Abstract

Downy mildew of peas is caused by the obligate parasite *Peronospora pisi*, which occurs sporadically throughout temperate pea-growing regions across the world. To screen pea lines against this biotrophic pathogen, a suitable and reproducible in vitro method using living plant material is required. Field screening can be influenced by environmental factors, thus giving variable results. The aim of this study was to develop a method that could reliably be used to screen pea cultivars against *P. pisi* in a laboratory setting. A range of bioassays were used to test various methods of inoculation, utilizing sporangia and naturally infested soil. Latent infection was achieved by planting seeds in soil collected from a site with a known history of *P. pisi* infection and directly inoculating young pea plants with sporangia. Out of the 108 plants which survived the experimental period, only two plants expressed visible signs of disease; however, through a two-step nested PCR process we detected latent infection in 24 plants. This research highlights the importance of considering the presence of latent infection when screening pea lines against downy mildew.

## 1. Introduction

Downy mildew disases are caused by Oomycete parasites that can infect and cause damage to a multitude of horticultural and ornamental plants. *Peronospora viciae* f. sp. *pisi* (syn. *P. pisi*) is the causal organism of downy mildew disease on peas (*Pisum sativum*) [1]. Infected plants may present as symptomatic with signs of disease, expressing sporangia on the underside of the leaves and stems. Severe infections can result in overall plant distortion, which may lead to the early death of young plants [2]. *Peronospora pisi* will often go undetected, remaining asymptomatic within the plants until a 12-h period of at least 90% relative humidity creates a conducive environment for disease expression [3]. This can result in sporadic outbreaks with the disease severity and expression being dependent on local environmental influences, such as the climate and weather, the mode of initial infection and the agricultural practices in the cropping area [2].

Fungicides are often utilized to manage outbreaks; however, as downy mildew are Oomycetes and not ‘true’ fungi, they tend to be susceptible to a narrow range of fungicides. In New Zealand, a study conducted by Harvey (2003) [4] concluded that the use of foliar fungicides was economically unviable, whereas Falloon et al. (2000) [5] confirmed a natural field resistance to metalaxyl-based seed treatments, with 56% of the New Zealand isolates tested having a notable level of insensitivity to phenylamide fungicides. Breeding pea lines for resistance towards *P. pisi* has occurred in New Zealand since 1980 [6]. Differential cultivars are often screened against natural populations of *P. pisi*; such studies have identified virulence variations and physiological specialization between isolates, leading to informal descriptions of *P. pisi* pathotypes [7,8,9]. Often cultivars are assessed under field conditions; however, the innate variability and low reproducibility associated with field testing means a controlled lab screening method is desired. 

Biotrophic organisms, such as downy mildews, are difficult to maintain under laboratory conditions, and can be maintained on living host plants through successive transfers between infected and healthy tissues. Pea plants are most commonly inoculated with *P. pisi* using a spore suspension [7,8,10,11,12], and success has been reported using naturally infested soil [9]; however, these studies do not consider the potential for asymptomatic infection. Thus, the aim of this technical study was to evaluate bioassay techniques that would enable the screening of pea plants against *P. pisi* in a controlled laboratory setting.

## 2. Results

### 2.1. Fresh Sporangia Assays

The use of fresh sporangia in Experiments 1–4 resulted in irregular asymptomatic infection; however, plants did not express signs of disease in any of the sporangial assays (Table 1). One seed did not germinate in Experiment 1 and molecular analysis indicated asymptomatic infection in three out of the nine remaining plants. Two-step nested PCR indicated asymptomatic infection in seven out of 10 plants from Experiment 2. Of the six seeds from Experiment 3, only one germinated and survived the 6-week growth period, which nested PCR later indicated to be asymptomatically infected. Five out of the six seeds from Experiment 4 germinated and all were identified as having asymptomatic infection upon nested PCR analysis.

### 2.2. Soil Experiments

During Experiment 5, 50 seeds were grown in soil collected from Site Two. Fifteen plants survived the 6-week growth period, whereas the nested PCR analysis and subsequent sequencing identified asymptomatic infection in seven plants. Thirty-five plants either did not emerge or died 2–3 days after emergence. Of the 93 seeds sown in Experiment 6, 68 germinated and survived the 6-week growth period: 27/36 seeds sown in soil from Site One, 33/48 seeds sown in soil from Site Two and 9/9 seeds sown in soil from Site Three (Table 1). Upon molecular analysis, one plant was identified as having asymptomatic infection. Visual inspection identified *P. pisi* sporulation on two plants (‘Utrillo’ and ‘CS480-AF’) and identity was confirmed after DNA sequencing. A BLAST against a known sequence of a *P. viciae* isolate obtained from a *Pi. sativum* plant (accession AY225471) indicated 99% identity, with only two nucleotide substitutions. The sequence was deposited in NCBI with the accession number MW623429. Tissue taken from an asymptomatic leaf, adjacent to one which was sporulating, was not found to have asymptomatic infection.

## 3. Discussion

The overall aim of this research was to investigate possible preliminary bioassay techniques to enable the successful screening of pea plants for resistance against *P. pisi* in a controlled laboratory setting. The use of fresh sporangia and naturally infested soil resulted in asymptomatic infection of *Pi. Sativum* plants; however, limited disease expression was observed, with sporulation only seen on two plants across all experiments in this study.

Maintaining a biotrophic pathogen under laboratory conditions often requires successive inoculations using fresh sporangia between susceptible hosts [12]. In the current study, the use of fresh sporangia resulted in asymptomatic infection in sixteen plants across Experiments 1–4. To directly expose young plants to *P. pisi*, sporangia was either placed on the abaxial side of the youngest leaf (Experiment 1) or on the apical tip (Experiment 2). PCR results indicated that inoculating the apical tip was more successful in causing asymptomatic *P. pisi* infection; 70% plants from Experiment 2 produced a positive band compared with 30% from Experiment 1. Due to sporangia being placed directly on the plant, either on a leaf or on the apical tip, it cannot be entirely excluded that remnant sporangia were detected in PCR in place of asymptomatic infection; however, this is unlikely due to the growth of the plant over the trial period. The authors suggest that future work involving in planta DNA extraction should consider surface sterilization treatments, followed by treatment with a DNA chelating agent, such as propidium monoazide, to exclude the amplification of relic DNA [13,14]. In contrast to methods used in the current study, Mence and Pegg (1971) [15] achieved symptomatic, systemic infection by pipetting a sporangial suspension over the apical region of their plants. In the same study, Mence and Pegg (1971) [15] examined the effect of the inoculation site on systemic infection and reported that sporangial suspensions are more effective when pipetted into the bud rather than the youngest leaf lamina (60% became systemically infected when inoculated into the bud, compared with 2.5% when inoculated on the lamina), likely due to the increased susceptibility of younger tissues [16]. It is important to note that although the efficacy of inoculating the tip aligns with the current study, Mence and Pegg (1971) [15] achieved disease expression, which allowed them to visually assess the plants for signs and symptoms of downy mildew disease. This was not possible in the current study due to the lack of disease expression, likely due to insufficient humidity.

*Pisum sativum* seeds were also exposed to *P. pisi* sporangia in Experiments 3 and 4, with seeds in Experiment 3 having been soaked in a sporangial suspension for 24 h prior to being sown alongside a cluster of sporangia. Soaking seeds for 24 h in a suspension prior to being sown appeared to affect the germinability of the seeds; only one out of six seeds germinated and survived the experimental period from Experiment 3, whereas five of the six seeds appertaining to Experiment 4 germinated and survived. The reason for this contrast in seed survival is unknown and can only be speculated; however, it is conceivable that unknown, infective pathogens were also present on the infected tissue, or the seed succumbed to the *P. pisi* inoculum load. Previous studies [17,18,19] germinated seeds for 3 days on either filter paper or agar, prior to exposing the radicle to inoculum. In the current study, seeds were not pregerminated to reduce the chance of potentially exposing the radicle to secondary infection and simulating conditions in which a seed could become exposed to inoculum in the environment. Hickey and Coffey (1977) [17] germinated seeds on dampened filter paper for three days, then immersed them in a sporangia suspension for 30 min prior to sowing. Comparatively, the seeds in the current study were soaked for considerably longer (24 h), perhaps overwhelming the seed with infective inoculum and leading to the poor survival rate. Ryan (1971) [18] investigated the effect of the length of the immersion time on inoculation success and noted that seeds which had been pregerminated for 48 h resulted in 90–100% infection when immersed in a sporangial suspension for 15–20 min, whereas pregerminated seeds immersed for 1 min had an infection rate of only 58%. Upon comparison of the current study and the 1977 [17] and 1971 [18] studies, it appears that immersing a dry seed for a longer period (24 h) will not result in a similar effect to immersing a pregerminated seed for a relatively short time (20 min). Wiesel et al. (2016) [19] employed a similar technique to that utilized in Experiment 4, in that the seed is sown alongside infective sporangia; however, seeds were pregerminated on agar and the sporangia was placed on the exposed radicle during sowing. Wiesel et al. (2016) [19] do not report on the success of their inoculation and infection rates directly; however, they noted *P. pisi* disease on 4–52% of plants, respective of cultivar. Despite the lack of disease expression, nested PCR indicated asymptomatic infection in 100% of plants in Experiment 4, reflective of this method’s efficacy in causing *P. pisi* infection in pea plants. Although Wiesel et al. (2019) [19] tested cultivars of various resistances and susceptibilities to *P. pisi*, it is unknown if any of their symptomless plants were hosting latent infection.

Experiments 5 and 6 investigated naturally infested soil as an avenue for infecting pea plants with *P. pisi*, using soil collected from three different sites. Fifteen of the fifty seeds sown in Experiment 5 survived the 6-week growth period. Nested PCR analysis indicated seven out of the fifteen plants to be asymptomatically infected with *P. pisi*. In contrast, molecular analysis identified seven plants from Experiment 6 as being asymptomatically infected with *P. pisi*. It is possible that only seven plants from Experiment 6 were found to be positive for *P. pisi* due the molecular analysis having been completed in 2021; two years after the DNA had been extracted. Results obtained in previous research indicated that 111 sporangia or mycelial fragments containing nuclei are required in a plant/pathogen mixed DNA sample to be detected through the standard PCR process [20]. The PCR reaction utilized may not be sensitive enough to amplify infective propagules in planta. In addition, two plants expressed signs of disease (‘Utrillo’ and ‘CS480-AF’), although PCR on the examined piece of tissue from the same plants did not indicate infection. This inconsistency was investigated further by taking two separate samples from the ‘Utrillo’ plant; one of pure sporangia and the other from a section of an adjacent non-symptomatic leaf. The sporangial sample was positive for *P. pisi,* but the non-symptomatic leaf sample was not. Possible reasons for this discrepancy include that the ratio of *P. pisi* DNA to plant material was too low to detect, or the pathogen simply was not present in the sampled leaf section [17]. Hearth Mudiyanselage (2015) [21] reported inconsistencies between PCR and fluorescent microscopy results when examining *P. sparsa* infection locations within boysenberry plants, indicating the discontinuous colonization of plant tissue. They recommend assessing replicate plant tissue using both microscopy and PCR to overcome the variability in detection. 

Sixty-eight seeds germinated and survived the trial period (73.1%) in Experiment 6. Using naturally infested soil as a means of inoculating plants often raises concerns as other pathogens may be present, such as *Aphanomyces euteiches* or *Pythium* spp., which can influence seed germination and emergence. Stegmark (1991) [22] also reported low frequencies of infected seedlings after sowing seeds in naturally infested soil; however, plants in the reported study were noted as symptomatic with sporulation. As in the current study, Stegmark (1991) [22] incubated plants at temperatures between 12 °C and 15 °C and at 100% relative humidity. The success of using soil as a natural source of inoculum to bulk sporangia and achieve *P. pisi* infection was reported by Davidson et al. (2011) [9]. Comparatively, Davidson et al. (2011) [9] placed pea seeds in a tray with potting mix and added a 2-cm layer of infested soil on top of the seeds. This slight difference in methodology between the 1991 and 2011 studies and the current study is unlikely to explain the differences in disease expression in the current study. Potentially, a lack of viable inoculum in the soil could have contributed to the lack of disease expression. Davidson et al. (2011) [9] and Stegmark (1991) [22] did not report on the quantity of inoculum in the soil used in their research; therefore, comparisons cannot be made between these and the current study.

The lack of disease symptoms and confirmations of asymptomatic infection by nested PCR indicates a period of latency during this study. The confirmation of asymptomatic infection with PCR and the reported success of experiment variations within the literature suggest that the inoculation techniques utilized in this study could be used to achieve infection and disease expression. Further experiments would need to be conducted to identify the incubation conditions required to break latency and cause disease, and whether the choice of cultivar impaired the pathogen’s ability to cause symptoms. Disease expression occurred when the temperature was maintained at 15 °C throughout a 16-h photoperiod, akin to the conditions reported by Wiesel et al. (2016) [19]. The temperatures at which inoculated plants were incubated to induce infection and disease expression vary between studies; however, the conditions tested in this study either fall within or replicate the suggested parameters [3,8,10]. Regardless of temperature variations in the previously published literature, the need for high humidity remains absolute [2,3,10,23]. High humidity was managed by incubating the plants within plastic bags. Pegg and Mence (1970) [3] investigated the effect of relative humidity (rh) further, and noted that *P. pisi* sporangial density on pea plants was highest at 100% rh and decreased entirely below 91% rh. As conditions inside the bags were not measured, it is unknown if the plants were subjected to the intended conditions; humidity over 91% and temperatures remained between 8 °C and 17 °C. Latency periods are commonly observed in pathogen–host interactions, such as wheat powdery mildew (*Blumaeria graminis* f. sp. *tritici*), *Botrytis cinerea* and basil and boysenberry downy mildews (*Ocimum basilisum* and *Peronospora sparsa*) [21,23,24,25]. Often, field infections of *P. pisi* are symptomless until environmental conditions trigger sporulation, indicating a latency period between initial infection and sporulation [26]. Additional research needs to be conducted to assess all laboratory conditions relating to *P. pisi* disease expression in order for more accurate conclusions to be drawn. 

## 4. Materials and Methods

Six experiments were undertaken to assess various methods of inoculating pea plants with *P. pisi*. Two sources of inoculum were utilized: sporangia and naturally infested soil. Inoculum was obtained from three sites in Rakaia, Canterbury, New Zealand (Site One: 43°45′38.56″ S, 171°57′47.50″ E; Site Two: 43°45′35.98″ S, 171°57′52.66″ E; and Site Three: 43°45′23.29″ S, 171°58’18.40″ E).

### 4.1. Inoculum Collection and Preparation

#### 4.1.1. Sporangia

Fresh sporangia were collected from a single, highly diseased, volunteer *Pi. sativum* ‘Utrillo’ plant at Site Two in June 2019. Diseased plant material was placed into paper bags for transportation. After morphological identification using stereo and compound microscopes, sporangia were immediately utilized for experiments.

#### 4.1.2. Soil

Naturally infested soil was collected from Site Two in June 2019 for use in Experiment 5. Twenty-five samples were collected in a grid formation: 1 m^2^ apart within a 5 m^2^ plot. For each of the 25 samples, four soil cores were taken from the top 10 cm of soil, using a soil corer (25 mm diameter), and combined in plastic bags. Soil collected for Experiment 6 was collected from Sites One, Two and Three in July 2019. Each site was sampled by walking from East to West in a ‘W’ formation. Thirty-one samples were collected in total; twelve samples were collected from Site One, sixteen samples from Site Two and three samples from Site Three. The number of samples collected from each site was representative of the size of the sampling area. As previously described, each sample consisted of four cores, which were combined in a plastic bag. Stones and other large debris were removed from all samples, and the remainder of the sample was homogenized. Samples were kept at 4 °C until required.

### 4.2. Experimental Design

#### 4.2.1. Fresh Sporangial Inoculation Experiments

To prepare for Experiments 1 and 2, 20 *Pi. sativum* cv. ‘Utrillo’ seeds were sown into individual, 200-mL pots containing potting mix (4:1 bark to pumice media, Osmocote^®^ 3–4 month-controlled release fertilizer, lime and hydroflo) and grown in an incubator at 15 °C with a 16 h photoperiod, until the first three sets of leaves had formed—approximately 4 weeks. Experiment 1: Ten young *Pi. sativum* cv. ‘Utrillo’ plants were randomly selected and sprayed with RO water containing two drops of Tween 20 (LabChem) per 500 mL, until runoff. Clusters of sporangia (approximately 2 mm^2^) were lifted from the diseased field plant with a sterile needle. For each of the ten plants, one cluster was placed on the abaxial side of the youngest leaf. The soil was watered with RO water and plants were immediately placed into a plastic bag to maintain a humidity over 95%. Plants were incubated for 6 weeks in a cycle of 16 h light at 17 °C and 8 h dark at 15 °C. Experiment 2: The remaining 10 prepared plants were sprayed with RO water, containing Tween 20 (as previously described), until runoff. Another 10 clusters (each cluster approximately 2 mm^2^) of sporangia were lifted from the diseased field plant with a sterile needle; one cluster was placed on the apical tip of each plant. The soil was watered with RO water and the plants were immediately placed into plastic bags to maintain a high humidity. Plants were incubated under the same conditions as those of Experiment 1. Experiment 3: Six *P. sativum* ‘Utrillo’ seeds were soaked in a Petri dish that contained 25 mL of RO water and 3 pieces (3–6 mm^2^) of diseased leaf tissue with profuse sporulation for 24 h (16 h light at 17 °C and 8 h dark at 15 °C). As previously described, each seed was placed into individual pots containing potting mix. Prior to being covered with mix, a 5 mm^2^ sporangial cluster was placed on top of each seed. Pots were placed into a growth chamber for 6 weeks, under the same conditions as those of Experiments 1 and 2. Experiment 4: Six *P. sativum* ‘Utrillo’ seeds were placed into individual pots containing potting mix. Before being covered with mix, a 5 mm^2^ cluster of sporangia was placed on top of each seed. Pots were watered with RO water, placed into plastic bags and incubated for 6 weeks under the same conditions as those of Experiments 1, 2 and 3.

#### 4.2.2. Soil Inoculum Experiments

Experiment 5: Soil samples collected from Site Two were placed into 25 individual 200-mL seed-raising pots and had two ‘Utrillo’ seeds sown into each pot (50 seeds in total). The pots were watered with RO water until saturation and placed into a plastic bag. Over a 6-week period, plants were incubated in a cycle of 16 h light at 17 °C and 8 h dark at 15 °C. Experiment 6: Soil from the 31 samples was placed into individual, 200-mL seed-raising pots. One seed from each of the *P. sativum* cvs. ‘CS480-AF’, ‘Rondo’ and ‘Utrillo’ were sown into each pot; totaling three seeds per pot. The pots were watered to saturation and placed into plastic bags to maintain a high humidity, then incubated for 6 weeks at 15 °C with a 16 h photoperiod. Plastic bags were not removed during this period.

Throughout incubation, all plants were monitored for characteristic symptoms and signs of *P. pisi* structures, such as aseptate hyphae and sporangia. Plants were misted with RO water using a hand-held spray bottle as required; indicated by the formation of condensation in the inside of containers and plastic bags.

### 4.3. Molecular Detection of *Peronospora pisi*

#### 4.3.1. DNA Extraction

At the end of the trial period, a 5 mm^2^ piece of tissue was removed from the youngest, fully formed leaf of each plant, from each experiment. Plants with visible infection had an additional two samples taken; 3 mm^2^ of sporangia was lifted from the plant and an adjacent, asymptomatic leaf had a 5 mm^2^ piece dissected from its center. Samples were placed into individual tubes (1.7 mL) containing 300 µL aliquots of 10% Chelex 100 (BioRad). Tubes were vortexed for 2 sec, three times, then placed onto a block heater (Stuart SBH130D) for 10 min at 100 °C. Each tube was vortexed for another 2 sec, three times, and returned to the heat block for a further 10 min at 100 °C. Tubes were centrifuged for 10 min at 13,000 rpm. The supernatant (approximately 200 µL) was aliquoted into 0.6 mL tubes and stored at 4 °C.

#### 4.3.2. Two-Step Nested PCR

A two-step nested PCR was performed using forward primer DC6 (5′-GAGGGACTTTTGGGTAATCA-3′), specific to Peronosporales and Pythiales and universal reverse primer ITS4 (5′-TCCTCCGCTTATTGATATGC-3′) [27] in the primary reaction and *P. viciae*-specific primers DM3F (5′-GCCGAGTGAGCCCTATCATGGTGAGTGTT-3′) and DM3R (5′-TATGCTTAAGTTCAGCGGGTAATCTTGCCT-3′) [10] in the secondary reaction. The PCR product from the first reaction was used as the DNA template in the secondary reaction. Reactions were prepared in 20 µL volumes; 1 U DreamTaq Green PCR Master Mix, 0.125 µM forward and reverse primers, 0.5 µL of DNA template and the remaining volume with water. Template DNA from the primary reaction was diluted in nuclease-free H_2_O, at a ratio of 1:9, before being added to the secondary reaction. Non-template controls using nuclease-free water were included in each reaction. Cycling parameters for the primary reaction involved an initial denaturation of 95 °C for 2 min, 30 cycles of denaturing at 95 °C for 20 s, annealing at 55 °C, an extension at 72 °C for 72 s and a final extension at 72 °C for 10 min. The secondary reaction underwent an initial denaturation of 94 °C for 5 min, 40 cycles of denaturing at 94 °C for 30 sec, annealing at 55 °C for 1 min, an extension at 72 °C for 1 min and a final extension at 72 °C for 10 min.

#### 4.3.3. Visualization and Identity Confirmation

PCR products were separated on a 1.5% agarose gel precast with ethidium bromide (0.2 µM) for 45 min at 90 V, then visualized under UV light using GelDoc. To confirm the presence of *P. pisi* asymptomatic infection, the samples from each experiment that produced the brightest positive bands were selected and sequenced at the Lincoln University sequencing facility. Product sizes were expected to be approximately 1300 (DC6/ITS4) and 725 (DM3F/R) base pairs [10,27]. The sequences were viewed and edited using BioEdit (version 7.2.6.1) [28] and the basic local alignment search tool (BLAST) was used to confirm identity.

## 5. Conclusions

In this study, we aimed to identify a successful and reliable method for screening pea plants against *P. pisi* in a controlled setting. Asymptomatic infection was achieved by planting seeds in soil collected from a site with a known history of *P. pisi* infection, placing inoculum alongside the seed at the time of planting, and exposing young plants to fresh *P. pisi* sporangia. The variation observed between visual inspection and molecular analysis highlights the importance of considering the conditions in which peas are assessed for their tolerance to downy mildew. Field assessments are prone to natural variabilities, such as temperature and humidity, which may affect the number of diseased plants observed; analogous to the lack of sporulation throughout this research. As observed in this study, plants may host latent infections and remain symptomless until the conditions are optimal, leading to severe outbreaks at crucial periods, such as seed formation. This study highlights the need for the testing of symptomless plants when assessing a cultivar’s resistance potential.

## Figures and Tables

**Table 1 pathogens-10-01312-t001:** Comparison of the six experiments conducted in this study and the resulting number of plants which survived the trial period, expressed signs of disease, and produced a positive band after a two-step nested PCR, subsequent to exposure to *Peronospora pisi* inoculum. Initial *N* = number of seeds or plants exposed to inoculum in each experiment.

Experiment	Initial *N*	Inoculum Type	Survived Trial Period	Expressed Signs of Disease	Asymptomatic Infection Detected by PCR
1	10	Sporangia	9	0	3
2	10	Sporangia	10	0	7
3	6	Sporangia	1	0	1
4	6	Sporangia	5	0	5
5	50	Soil	15	0	7
6	93	Soil	68	2	1

## Data Availability

The datasets generated during and/or analyzed during the current study are all included in this manuscript.

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
