# Peer review of "Short Communication: Latent Detection of Downy Mildew (Peronospora pisi) in Bioassays against Pisum sativum"

_pathogens, 2021, doi:10.3390/pathogens10101312_

Round 1

Reviewer 1 Report

The reason given for this paper is to create a suitable and reproducible in vitro bioassay to screen pea lines against downy mildew because field trials are too variable. This will be of interest to researchers and breeders. Different, highly appropriate inoculation methods using sporangia or infested soil were tested. But previously published methods don’t seem to have been used for comparative purposes which could have strengthened the paper. However, two methods did give a high incidence of what appears to be symptomless or latent infection.  Issues arising are whether the journal has a policy to require a repeat of this kind of experiment to confirm results or not. Secondly, there is a question as to why there was mainly only symptomless infection if conditions the plants were grown in were ideal for allowing sporulation? Certainly a repeat of the two most successful methods (Exp 2 and 4) would confirm the methods work and also allow some verification of whether symptoms could be induced or whether infection was only symptomless, or if the PCR method had detected remnants of the inoculum. In addition there are some other edits required to improve the paper:

L18 – suggest delete ‘using’

L40 – suggest changing ‘natural resistance’ to ‘insensitivity to phenylamide fungicides’.

L84 – suggest adding ‘for resistance’ after “…pea plants…”

L87-89 – please add brief details of the pod-based inoculum method including the source and method to the methods section and results.

L92 - nineteen plants was 16 according to Table 1

L95-95 & 96-97 Experiments 6 and 7 were actually given as Exps 1 and 2 in the methods

L120 – “The reason for this contrast in seed survival is unknown and can only be speculated” – well it seems highly likely that either the downy mildew inoculum killed the seeds or another pathogen on the infected leaves may have been the problem. A 24h soak wouldn’t normally kill dry seeds as it takes that long to imbibe but maybe the soil was also too wet? Either way it makes this particular method (treatment 3) too unreliable. So that is a clear conclusion to take.

L121-129 Why weren’t the methods of references 16, 18 & 19 tested in this study as a comparison ? Can the authors discuss how the success rate of their best methods (Exp2 & 4) compare to these earlier studies?

L176-179 has no previous work been done to find out what conditions lead to sporulation rather than symptomless infection? Reference 3 cited in lines 30-31  suggests that some of this has been studied. Can the authors comment further rather than say simply that nothing can be concluded?

Text on lines 125-127 says “As successive replicates of Experiment 3 were not conducted, conclusions cannot be drawn on the possible effect of soaking seeds in water containing sporulating leaf material prior to sowing. “  Does this mean that there were repeats made of the other treatments (or experiments)? If so, the results should be presented.

Conclusions – “Due to the variation observed between visual inspection and molecular analysis, conclusions drawn from this study should be tentative”  Exp 4 had 5 out of 5 plants that germinated successfully infected albeit symptomlessly, and Exp 2 had 7/10 successful infections. The method used in treatment 4 was the same as treatment 3 (apart from Trt 3 also getting washed in inoculated water for 24h), but 5/6 seeds didn’t germinate, presumably these had rotted well before the end of the experiment and could not be tested by PCR.  Overall, this suggests some variability may be due to the source of sporangia used in each individual inoculation but essentially methods used in treatments 2 and 4 were sufficient if the symptomless infection is real rather than remnant inoculum.

The second key point (line 113 and conclusions) is why only symptomless infection occurred rather than sporulation if plants were incubated in ideal conditions for sporulation?  The authors suggest that it is unlikely that remnants of the inoculum had been detected but this does not explain why symptoms only appeared in two plants out of all the treatments/experiments.

Finally – if latent infection was confirmed - can latent infection also explain why the field plot tests were so variable to merit the in-vitro development?  Would it simply be better to wait or create the correct conditions for sporulation for the full symptoms to develop rather than measure latent infection with a PCR test?

Reviewer 2 Report

Authors have made the necessary improvements.

Author Response

The authors thank the reviewer for taking the time to review the paper. All necessary changes suggested by all reviewers have been undertaken.

Round 2

Reviewer 1 Report

I am satisfied that edits made in the latest version of the article address all my points. I am assuming that the journal does not require a repeat of experiments as this is a short communication and because there were closely related or similar experiments that reinforce each other, I am confident that the conclusions made are merited from the data.

This manuscript is a resubmission of an earlier submission. The following is a list of the peer review reports and author responses from that submission.

Round 1

Reviewer 1 Report

This manuscript aims to develop a screening method that could be used reliably to screen pea cultivars against P. Pisi in a laboratory setting.

I have the following comments for this study.

First, this study cannot be submitted as an Article. It is a highly technical assay with very limited data, so it should be submitted as a Note.

The title is not accurate. According to the data and results, the authors were not able to develop a bioassay for downy mildew against Pisum sativum.

I believe that this study is insufficient because of the limited number of plants used in the experiments. Experiment no. 1 used a total of 9 plants, while Experiment no. 11 used a total of 93 plants, as shown in Table 1. The limited number of plants used in the experiments makes it very difficult to assess the validity of the data.

Moreover, the data are not conclusive enough in order to be used for the development of an accurate bioassay for downy mildew. There was variation in the material studied, i.e., fresh material versus dried, there was variation between temperature and humidity control, and there was variation between visual inspection and molecular analysis, suggesting that any conclusions are vague.

Reviewer 2 Report

The manuscript pathogens-1187938 baring the title 'Bioassay development for downy mildew (Peronospora pisi) screening against Pisum sativum' by the authors Alexia Marr, Seona Casonato, Eirian Jones is an interesting description of research aimed at pea infection with the downy mildew pathogen. The authors describe very well the various attempts of infection but essentially failed to get visible and viable disease. They concluded that the attempts of infection finally achieved latent infection.

This study aimed to identify a successful and reliable method for screening pea plants against P. pisi but essentially screening was not demonstrated. Asymptomatic infection was achieved by planting seeds in soil with a known history of P. pisi infection. The two latter methods tested dehydrated inoculum, but infection did not occur. Due to the variation observed between visual inspection and molecular analysis, conclusions drawn from this study are essentially focused only in molecular detection of latent infection. I recommend minor changes as describe below. Yet the manuscript should be published as short communication.

Recommended changes:

Title

Change so it will read: 'Development of latent downy mildew (P p…) detection in Pisum sativum' because screening was not demonstrated and only latent infection was achieved and described using the PCR method.

Abstract

Shorten the abstract. I believe that the 200 hundred words limit is sufficient.

Text

L 37 Abbreviate Peronospora to P. here and elsewhere, where possible.

L 62-63 a cultivar or cultivars or a cultivar's

Comma are missing in several places lile Line 78 , therefore,…  Line 120 ' however,… L 206 , therefore, … L 211 and line 219 , however,… and perhaps elsewhere.

L 160 since exp was not repeated, omit it from list of exps and only make a note in the text.

L 289-290 Omit table and include the sites in the text.